# Novelties on Neuroinflammation in Alzheimer’s Disease–Focus on Gut and Oral Microbiota Involvement

**DOI:** 10.3390/ijms252011272

**Published:** 2024-10-19

**Authors:** Cristina Popescu, Constantin Munteanu, Aurelian Anghelescu, Vlad Ciobanu, Aura Spînu, Ioana Andone, Mihaela Mandu, Roxana Bistriceanu, Mihai Băilă, Ruxandra-Luciana Postoiu, Andreea-Iulia Vlădulescu-Trandafir, Sebastian Giuvara, Alin-Daniel Malaelea, Gelu Onose

**Affiliations:** 1Faculty of Medicine, University of Medicine and Pharmacy “Carol Davila”, 020022 Bucharest, Romania; cristina_popescu@umfcd.ro (C.P.); aurelian.anghelescu@umfcd.ro (A.A.); aura.spinu@umfcd.ro (A.S.); ioana.andone@umfcd.ro (I.A.); roxana.bistriceanu@rez.umfcd.ro (R.B.); mihai.baila@umfcd.ro (M.B.); ruxandra-luciana.postoiu@drd.umfcd.ro (R.-L.P.); andreea-iulia.trandafir@drd.umfcd.ro (A.-I.V.-T.); sebastian.giuvara@gmail.com (S.G.); alin.malaelea@gmail.com (A.-D.M.); gelu.onose@umfcd.ro (G.O.); 2Neuromuscular Rehabilitation Clinic Division, Clinical Emergency Hospital “Bagdasar-Arseni”, 041915 Bucharest, Romania; 3Department of Biomedical Sciences, Faculty of Medical Bioengineering, University of Medicine and Pharmacy “Grigore T. Popa” Iași, 700454 Iași, Romania; 4Department of Computer Science and Engineering, Faculty for Automatic Control and Computers, University Politehnica of Bucharest, 060042 Bucharest, Romania; vlad.ciobanu@cs.pub.ro

**Keywords:** Alzheimer’s disease (AD), neuroinflammation, gut microbiota, dysbiosis, amyloid peptides

## Abstract

Recent studies underscore the role of gut and oral microbiota in influencing neuroinflammation through the microbiota–gut–brain axis, including in Alzheimer’s disease (AD). This review aims to provide a comprehensive synthesis of recent findings on the involvement of gut and oral microbiota in the neuroinflammatory processes associated with AD, emphasizing novel insights and therapeutic implications. This review reveals that dysbiosis in AD patients’ gut and oral microbiota is linked to heightened peripheral and central inflammatory responses. Specific bacterial taxa, such as Bacteroides and Firmicutes in the gut, as well as Porphyromonas gingivalis in the oral cavity, are notably altered in AD, leading to significant changes in microglial activation and cytokine production. Gut microbiota alterations are associated with increased intestinal permeability, facilitating the translocation of endotoxins like lipopolysaccharides (LPS) into the bloodstream and exacerbating neuroinflammation by activating the brain’s toll-like receptor 4 (TLR4) pathways. Furthermore, microbiota-derived metabolites, including short-chain fatty acids (SCFAs) and amyloid peptides, can cross the blood-brain barrier and modulate neuroinflammatory responses. While microbial amyloids may contribute to amyloid-beta aggregation in the brain, certain SCFAs like butyrate exhibit anti-inflammatory properties, suggesting a potential therapeutic avenue to mitigate neuroinflammation. This review not only highlights the critical role of microbiota in AD pathology but also offers a ray of hope by suggesting that modulating gut and oral microbiota could represent a novel therapeutic strategy for reducing neuroinflammation and slowing disease progression.

## 1. Introduction

Alzheimer’s disease (AD) is a progressive neurodegenerative disorder [1], primarily characterized by memory loss, cognitive decline, and behavioral changes [2]. As the most prevalent form of dementia, AD accounts for 60–80% of all dementia cases worldwide [3]. Currently, over 50 million people globally live with dementia, with AD being the leading cause. The risk of developing AD increases significantly with age, doubling approximately every five years after the age of 65. Nearly half of individuals aged 85 and older have AD, with prevalence rates reaching about 25–30%. This trend is expected to continue as the global population ages, posing significant challenges for healthcare systems and society [4].

The societal and economic impact of AD is profound, affecting not just those diagnosed but also their families, caregivers, and the broader healthcare system. As the number of older adults increases worldwide, the prevalence of AD is projected to rise dramatically, escalating the demand for healthcare services, long-term care, and economic resources [5]. The psychosocial effects are also considerable, with patients experiencing a gradual loss of independence and quality of life. They face increasing difficulties in communication, daily activities, and social interactions. Meanwhile, caregivers often endure significant emotional and physical stress, financial burdens, and reduced quality of life due to the high demands of caregiving. This scenario underscores the need for comprehensive strategies to manage the growing impact of AD on individuals and communities [6].

A combination of genetic, environmental, and lifestyle factors influences AD prevalence [7,8]. High-income countries with older populations, such as those in North America and Western Europe, report higher prevalence rates of AD. In contrast, lower prevalence rates are typically observed in low- and middle-income countries, although these rates rise rapidly as life expectancy improves and populations age [9]. Genetic factors play a crucial role in the risk of developing AD. The apolipoprotein E (APOE) ε4 allele is the most significant genetic risk factor for late-onset AD. Individuals carrying one copy of the APOE ε4 allele have an increased risk, while those with two copies face an even higher risk [10]. Additionally, early-onset AD, which occurs before age 65, is often linked to mutations in the APP, PSEN1, and PSEN2 genes, highlighting the genetic role [11].

Emerging research has increasingly focused on the role of neuroinflammation in AD, highlighting the significant contribution of inflammatory processes to disease progression [12]. Neuroinflammation in AD involves activating glial cells, particularly microglia and astrocytes, the brain’s resident immune cells. Initially, microglia attempt to clear amyloid-beta (Aβ) plaques through phagocytosis, but chronic activation releases pro-inflammatory cytokines and chemokines, exacerbating neuronal damage. Astrocytes, which support neuronal function and maintain the blood-brain barrier (BBB), also contribute to the inflammatory milieu by releasing inflammatory mediators. This persistent activation of glial cells creates a cycle of inflammation and neurodegeneration, further accelerating AD progression [13].

In the elderly population, a significant shift in gut microbiota composition, known as dysbiosis, is frequently observed, characterized by a decline in microbial diversity and an increase in pathogenic species [14]. Studies have shown that this imbalance often leads to an overrepresentation of Gram-negative bacteria, such as Escherichia coli and Bacteroides. These microorganisms produce lipopolysaccharides (LPS), potent pro-inflammatory endotoxins that can exacerbate systemic inflammation, a critical contributor to the pathogenesis of AD [15].

Aging is characterized by the gradual decline of physiological functions, including reduced cellular repair and tissue renewal capacity, diminished resilience to stressors, impaired immune function, and reduced metabolic efficiency, often resulting in chronic conditions and frailty [16]. As mentioned above, aging is associated with a marked decline in immune function, particularly cellular immunity [17]. Immunosenescence affects the adaptive and innate immune systems, compromising the body’s ability to respond effectively to pathogens, including bacteria and fungi [18]. Reduced T-cell activity and impaired phagocytic functions in macrophages and microglia exacerbate this vulnerability, allowing for unchecked microbial proliferation, including Gram-negative bacteria and opportunistic fungal infections such as candidiasis [19].

Oral health issues, including dehydration, candidiasis (oral thrush), and poor oral hygiene, are common in elderly individuals with AD and can significantly impact both systemic and cognitive health [20]. Dehydration [21], often due to cognitive decline and difficulty swallowing, leads to dry mouth (xerostomia), fostering bacterial and fungal overgrowth, which increases the risk of infections like candidiasis. This fungal infection, common in elderly AD patients, is exacerbated by reduced immune function, medication side effects, and poor oral hygiene. Chronic oral infections and poor oral status can elevate systemic inflammation, which is linked to neuroinflammation in AD [22]. Pro-inflammatory molecules and bacteria from the mouth can enter the bloodstream, cross the BBB, and activate microglial cells, contributing to the accumulation of amyloid-beta plaques and tau tangles, hallmarks of AD. Thus, maintaining oral hygiene and hydration is crucial for reducing inflammation and potentially slowing AD progression. Proper dental care and infection management can improve quality of life. They may positively impact the course of AD by mitigating some of the inflammatory processes driving neurodegeneration [23].

Chronic constipation, common in older adults, is often linked to a dysregulated gut microbiota, characterized by a reduction in beneficial bacterial species, such as those producing short-chain fatty acids (SCFAs), and an increase in pathogenic or pro-inflammatory microorganisms. The role of gut microbiota in elderly constipation and its relevance to neuroinflammation highlight the importance of maintaining a healthy microbiome as a potential therapeutic avenue for mitigating AD progression. Targeting gut dysbiosis through dietary interventions or probiotics may offer promising strategies to reduce gastrointestinal and neuroinflammatory burdens in aging [24].

In individuals with AD, the weakening of these immune defenses leads to chronic infections, which further aggravates systemic inflammation and, in a vicious circle, disrupts immune regulation. The heightened systemic inflammatory state—low-grade inflammation—may result in peripheral immune cells infiltrating the central nervous system (CNS) through compromised BBB functioning, further contributing to neuroinflammation. This interplay between weakened immunity and microbiota-induced inflammation accelerates the progression of neurodegenerative changes in AD [25].

Recent studies have emphasized the role of the gut–brain axis and the involvement of gut microbiota in AD, revealing a novel aspect of neuroinflammation [26]. The gut–brain axis refers to the bidirectional communication network between the gastrointestinal tract and the central nervous system [27]. Dysbiosis, or an imbalance in gut microbiota composition, has been implicated in neuroinflammation and could contribute to the pathogenesis of AD [28]. Changes in the gut microbiota can affect the production of metabolites, modulation of the immune system, and gut permeability, all of which can influence brain health [29]. This connection suggests that gut microbiota may play a role in the inflammatory processes observed in AD, highlighting the potential for microbiota-targeted therapies [30].

The impact of normal gut microbiota on neuroinflammation and AD is further emphasized by the production of microbial metabolites, such as SCFAs, which have been shown to modulate inflammation and immune responses [31]. SCFAs, including butyrate, acetate, and propionate, are produced by the fermentation of dietary fibers by gut bacteria, can cross the BBB and influence microglial activity, and are shown to exert anti-inflammatory effects [32]. Conversely, dysbiotic gut microbiota can produce pro-inflammatory molecules, such as lipopolysaccharides (LPS), which have been linked to the activation of microglial cells and the triggering of Toll-like receptor 4 (TLR4) pathways, facilitating neuroinflammation and contributing to amyloid-beta (Aβ) peptide plaque accumulation. In older adults with AD, this microbial imbalance exacerbates the breakdown of the intestinal barrier, allowing LPS to translocate into circulation and reach the brain, intensifying inflammatory cascades that worsen neurodegeneration [33].

Despite decades of research, there is currently no cure for AD, and available treatments only provide symptomatic relief without altering the underlying disease progression. The complex and multifactorial nature of AD pathology poses significant challenges for developing effective therapies [34]. Recent research has increasingly focused on understanding the underlying mechanisms of the disease, including the roles of neuroinflammation, oxidative stress, and the gut–brain axis, to identify new therapeutic targets [35]. This research explores how these interconnected processes contribute to the initiation and progression of AD, intending to develop interventions that can modify the disease course [36].

One of the hallmark features of AD at the molecular level is the accumulation of amyloid-beta (Aβ) peptides in the brain, which form extracellular plaques [37]. Aβ peptides are derived from the cleavage of amyloid precursor protein (APP) by beta and gamma secretases [38]. The aggregation of Aβ into insoluble plaques disrupts cell communication and activates immune responses, contributing to neuroinflammation and neuronal damage [39]. Recent advances have been made in understanding the clearance of Aβ from the brain, highlighting the role of the perivascular drainage pathways and emphasizing how Aβ is cleared along the walls of blood vessels in the brain through a system of perivascular spaces. This process, also known as the glymphatic system, facilitates the removal of waste products, including Aβ, from the brain. However, when this clearance mechanism becomes impaired, Aβ can accumulate, contributing to plaque formation and disease progression [40].

Recently, the FDA-approved Eli Lilly’s drug Kisunla (formerly known as donanemab), an anti-amyloid therapy for AD, represents a significant step forward in treatment approaches. Unlike traditional therapies that primarily target symptoms, Kisunla focuses on modifying the underlying disease process by reducing amyloid-beta plaque accumulation in the brain, a key pathological hallmark of AD. Clinical trials have shown that Kisunla can slow cognitive and functional decline in patients with early-stage AD, offering new hope for altering the disease course rather than just managing its effects. This approval marks a significant advancement in the search for disease-modifying therapies in Alzheimer’s care.

Another significant pathological feature of AD is the presence of neurofibrillary tangles (NFTs), composed of hyperphosphorylated tau protein. Tau is a microtubule-associated protein that stabilizes neurons’ microtubules [41]. In AD, tau undergoes abnormal hyperphosphorylation, causing it to detach from microtubules and aggregate into tangles within neurons. This disrupts the transport system within neurons, leading to cell death and contributing to the neurodegenerative process [42].

The gut microbiota’s influence on systemic inflammation and neuroinflammation suggests that the gut–brain axis could be a critical factor in the disease’s progression. This hypothesis is supported by findings that individuals with AD often have altered gut microbiota compositions compared to healthy controls, with decreased levels of beneficial bacteria and increased levels of pro-inflammatory species. These changes in gut microbiota could promote neuroinflammation by affecting the permeability of the gut and BBB, allowing inflammatory mediators to enter the brain and exacerbate AD pathology [43].

Gasotransmitters, such as nitric oxide (NO), hydrogen sulfide (H_2_S), and carbon monoxide (CO), are involved in the pathophysiology of AD, linking redox balance, gut microbiota, and epigenetic processes [29]. NO, involved in vascular tone and neurotransmission, can exacerbate oxidative stress when dysregulated [44]. H_2_S is a potent antioxidant, supporting gut bacteria and modulating gene expression through histone acetylation [45]. CO provides anti-inflammatory effects, helps maintain gut barrier integrity, and influences epigenetic changes [46].

This comprehensive synthesis of AD research underscores this disorder’s complexity and multifactorial nature. It highlights the need for a holistic approach to AD research and treatment, emphasizing the importance of addressing the diverse factors contributing to disease progression. Researchers hope to develop more effective strategies for preventing, diagnosing, and treating AD by exploring the interplay between genetic, environmental, lifestyle, and biological factors.

The primary objective of this review is to compile and synthesize current research on the role of neuroinflammation and microbiota involvement in AD. The review employs a detailed and structured approach for searching, selecting, and analyzing relevant studies published between 1 January 2021 and 31 December 2023. A comprehensive literature search was conducted across multiple electronic databases, including PubMed, Web of Science, Scopus, and PEDro, to ensure a wide collection of relevant studies to support our work. The search terms were carefully chosen to capture studies focusing on neuroinflammation, microbiota, and AD (Table 1). 

A narrative synthesis was conducted to summarize the findings from the included studies, focusing on the role of neuroinflammation and microbiota in Alzheimer’s disease. This qualitative synthesis provided a descriptive overview of the evidence, highlighting the studies’ patterns, similarities, and discrepancies. 

## 2. Neuroinflammation in Alzheimer’s Disease

Neuroinflammation, a critical response within the central nervous system (CNS), plays a dual role in both the protection and progression of various neurodegenerative diseases [47]. It is characterized by the activation of glial cells, such as microglia and astrocytes, and the release of pro-inflammatory cytokines, chemokines, and reactive oxygen species (ROS). While acute neuroinflammation can be beneficial, aiding in debris clearance and promoting repair following CNS injury, chronic neuroinflammation becomes deleterious, contributing to neuronal dysfunction, synaptic loss, and cognitive decline. This persistent inflammatory state is a hallmark of neurodegenerative diseases, where it exacerbates disease pathology and accelerates progression by sustaining a cycle of neuronal damage and glial activation [48].

Neuroinflammation is increasingly recognized as a critical component in the pathogenesis of AD [49]. It involves a complex interplay between the central nervous system’s (CNS) immune cells, particularly microglia and astrocytes, and various inflammatory mediators [50]. While neuroinflammation can initially serve a protective role, chronic and uncontrolled inflammation contributes to neuronal injury and accelerates AD progression [51].

Microglia are the resident immune cells of the CNS, analogous to macrophages in the peripheral immune system. They play essential roles in maintaining CNS homeostasis, including clearing debris, responding to injury, and modulating synaptic connections [52]. In AD, microglia are activated in response to amyloid-beta (Aβ) plaques and other pathological changes. Activated microglia can adopt various phenotypes, broadly classified as pro-inflammatory (M1) or anti-inflammatory/neuroprotective (M2) [53].

In the context of AD, microglia often exhibit a pro-inflammatory phenotype, characterized by the production of cytokines such as interleukin-1β (IL-1β), tumor necrosis factor-alpha (TNF-α), and interleukin-6 (IL-6). These cytokines contribute to a local inflammatory environment, exacerbating neuronal damage. Chronic microglia activation leads to sustained inflammation, oxidative stress, and further accumulation of Aβ, creating a vicious cycle that perpetuates neurodegeneration [54]. While microglia initially attempt to clear Aβ plaques through phagocytosis, their ability diminishes as the disease progresses. This impairment is linked to changes in microglial gene expression and function, further contributing to the buildup of Aβ and inflammation. The reduced efficiency of Aβ clearance by microglia is a key factor in the progression of AD pathology [55].

Astrocytes are star-shaped glial cells that support neuronal function, maintain the BBB, regulate blood flow, and modulate synaptic activity. In AD, astrocytes become reactive in response to Aβ plaques, neurofibrillary tangles, and inflammatory signals from activated microglia. Reactive astrocytes undergo morphological changes characterized by hypertrophy and upregulation of glial fibrillary acidic protein (GFAP) [56].

Reactive astrocytes contribute to the neuroinflammatory milieu by releasing cytokines, chemokines, and reactive oxygen species (ROS). They also produce complement proteins that can exacerbate synaptic dysfunction and neuronal injury. Additionally, astrocytes can influence the activation state of microglia, further amplifying the inflammatory response in the AD brain [57].

In AD, astrocytic dysfunction impairs their ability to support neuronal health and maintain homeostasis. This includes disrupted glutamate regulation, excitotoxicity, and impaired potassium buffering, contributing to neuronal hyperexcitability. The loss of astrocytic neuroprotective functions further accelerates neurodegeneration [58,59].

A range of cytokines and chemokines are elevated in the AD brain, contributing to neuroinflammation. Key pro-inflammatory cytokines include IL-1β, TNF-α, and IL-6, which can induce a cascade of inflammatory responses. Chemokines such as CCL2 (MCP-1) and CXCL10 attract immune cells to the site of inflammation, perpetuating the inflammatory cycle [60].

The complement system, part of the innate immune response, is also implicated in AD. Components of the complement cascade, such as C1q and C3, are upregulated in AD and localize to Aβ plaques and NFTs. The activation of the complement system can lead to the formation of membrane attack complexes and opsonization of neurons, promoting their clearance by microglia and contributing to neuroinflammation and cell death [61].

The NLRP3 inflammasome is a critical mediator of neuroinflammation in Alzheimer’s disease (AD), primarily found in microglia, the brain’s resident immune cells. In AD, the NLRP3 inflammasome is activated by pathological stimuli, such as amyloid-beta (Aβ) plaques and tau tangles. This activation involves a two-step process: priming, which upregulates inflammasome components, and subsequent activation, triggered by cellular stress signals like reactive oxygen species (ROS) and potassium efflux. Once activated, the NLRP3 inflammasome facilitates the cleavage of pro-caspase-1, producing pro-inflammatory cytokines IL-1β and IL-18. These cytokines contribute to a neuroinflammatory environment that exacerbates neuronal damage, promotes further Aβ deposition, and accelerates AD progression. Emerging evidence also links NLRP3 activation to gut dysbiosis, where microbial metabolites such as lipopolysaccharides (LPS) may enhance inflammasome activity in the brain. Given its role in driving inflammation and neuronal injury, the NLRP3 inflammasome represents a promising therapeutic target in AD, with ongoing research into inhibitors that could potentially mitigate neuroinflammation and slow disease progression [62].

Toll-like receptors (TLRs) are pattern recognition receptors that detect pathogen-associated molecular patterns (PAMPs) and damage-associated molecular patterns (DAMPs) [63]. In AD, TLRs are involved in recognizing Aβ and triggering inflammatory responses. TLR activation on microglia and astrocytes produces pro-inflammatory cytokines and chemokines, contributing to the overall inflammatory state in the AD brain [64].

The blood-brain barrier (BBB) is a selective barrier that protects the CNS from harmful substances while allowing the passage of essential nutrients. In AD, BBB integrity is compromised, which can result from chronic inflammation and endothelial cell dysfunction. This disruption allows peripheral immune cells and inflammatory mediators to enter the brain, exacerbating neuroinflammation. The leakage of blood-derived molecules, such as fibrinogen and albumin, into the brain parenchyma can further activate glial cells and perpetuate the inflammatory response. BBB disruption also impairs the clearance of Aβ and other toxic substances from the brain, contributing to the accumulation of pathological proteins and neuroinflammation [65].

Systemic inflammation, often observed in aging and various comorbid conditions, can influence neuroinflammation in AD. Elevated levels of pro-inflammatory cytokines in the periphery can cross the compromised BBB and activate glial cells, linking systemic and central inflammation. In AD, peripheral immune cells, including macrophages and lymphocytes, can infiltrate the CNS after BBB disruption. These cells can adopt pro-inflammatory roles, further contributing to the neuroinflammatory environment and neuronal injury [66].

Epigenetic changes, such as DNA methylation and histone acetylation, regulate the expression of inflammation and oxidative stress genes. For example, hypermethylation of antioxidant genes can reduce their expression, increasing susceptibility to oxidative damage. Understanding these mechanisms is crucial for developing strategies to mitigate neuroinflammation and oxidative stress, providing a more nuanced approach to treating neurodegenerative diseases [67].

## 3. Microbiota Involvement in Neuroinflammation and Alzheimer’s Disease

### 3.1. Human Microbiota’s Role in Alzheimer’s Disease

The human microbiota comprises trillions of microorganisms, including bacteria, viruses, fungi, and archaea, that reside on and within various body parts, such as the skin, mouth, and gastrointestinal (GI) tract [68]. The gut microbiota is the most densely populated and diverse microbial community, essential in maintaining human health [69].

The gut–brain axis (GBA) is a bidirectional communication network that links the central nervous system with the enteric nervous system (ENS) of the GI tract and encompasses multiple pathways, including neural, hormonal, immune, and metabolic routes [70]. The primary neural pathway of the GBA involves the vagus nerve, which transmits signals between the gut and the brain [71]. The ENS, often called the “second brain,” contains a vast network of neurons capable of independent function but also communicates extensively with the CNS through the vagus nerve and spinal cord [72].

The gut–brain axis is influenced by hormones and neuropeptides produced by the gut and other organs [73]. These include ghrelin, leptin, and peptide YY, which regulate appetite, mood, and cognitive functions [74]. The hypothalamic–pituitary–adrenal (HPA) axis is another crucial component, linking stress responses to gut function [75]. The gut microbiota can modulate the body’s stress response via the HPA axis [76]. Dysbiosis is associated with altered stress responses and has been implicated in the development of mood disorders and cognitive impairments [77].

The immune system serves as an important intermediary in the GBA [78]. Cytokines and other immune mediators produced in the gut can affect brain function and vice versa [79]. The integrity of the gut barrier and the BBB are critical in maintaining proper communication and preventing neuroinflammation [80]. The gut microbiota helps in the maturation and function of the immune system, affecting neuroinflammatory responses. Dysbiosis, or an imbalance in the gut microbiota, can lead to systemic inflammation, affecting brain health and contributing to neurological disorders [81].

Microbial metabolites, such as SCFAs, bile acids, and tryptophan metabolites, play significant roles in the GBA. These metabolites can indirectly cross the BBB or influence neuronal activity by modulating systemic inflammation and metabolic functions [82].

SCFAs produced by microbial fermentation of dietary fibers have neuroactive properties. Butyrate, in particular, has anti-inflammatory effects and supports the integrity of the BBB [83]. SCFAs can also influence microglial activation, impacting neuroinflammation [84].

The gut microbiota produces various neuromodulatory molecules, including neurotransmitters (e.g., serotonin, dopamine, and gamma-aminobutyric acid (GABA)) and other metabolites that influence brain function and behavior [85,86]. These neurotransmitters can influence mood, behavior, and cognitive functions [87]. For instance, approximately 90% of the body’s serotonin is produced in the gut [88].

Emerging evidence suggests that the gut microbiota is involved in the pathophysiology of AD through its impact on neuroinflammation [89]. Dysbiosis in AD patients has been linked to increased intestinal permeability, systemic inflammation, and altered production of neuroactive compounds, all of which can contribute to AD pathology [90]. Understanding the gut–brain axis and its influence on neuroinflammation provides a new perspective on AD and offers potential avenues for therapeutic interventions targeting the microbiota [91].

Commensal bacteria in the gut compete with pathogenic microorganisms for nutrients and adhesion sites on the gut mucosa, producing antimicrobial compounds that inhibit pathogen growth. This barrier function is essential in preventing infections and maintaining gut health [92]. The gut microbiota aids in digesting complex carbohydrates, fibers, and other dietary components not digested by human enzymes [93]. It ferments these substrates to produce SCFAs, such as acetate, propionate, and butyrate, which are crucial for colonic health and have systemic metabolic effects [94].

The gut microbiota synthesizes essential vitamins, such as vitamin K and some B vitamins, and produces bioactive compounds that can influence host physiology [95].

The gut microbiota plays a critical role in the development and function of the immune system. It helps to educate and regulate immune responses, maintaining a balance between tolerance and immunity. This regulation prevents excessive inflammatory responses and protects against pathogenic infections [96].

The involvement of the gut microbiota in neuroinflammation and AD presents promising therapeutic potential [97]. Modulating the gut microbiota through probiotics, prebiotics, and dietary interventions could offer a novel approach to mitigating neuroinflammation and slowing AD progression [98]. Probiotics, which are live beneficial bacteria, and prebiotics, which are non-digestible food components that promote the growth of beneficial bacteria, can help restore a healthy microbiota balance [99]. Additionally, dietary changes that promote gut health, such as increasing fiber intake and reducing consumption of processed foods, could positively impact brain health [100,101].

Dysbiosis, or the imbalance of gut microbiota, has been associated with increased intestinal permeability, often called “leaky gut” [102]. This condition allows for the translocation of bacterial endotoxins, such as lipopolysaccharides (LPS), into the bloodstream, promoting systemic inflammation. LPS and other microbial metabolites can cross the BBB, contributing to neuroinflammatory processes in the brain [103].

The gut microbiota also holds the potential for developing new biomarkers for early diagnosis and progression monitoring of AD. Specific microbial signatures associated with AD could be identified and used to detect the disease in its early stages, long before cognitive symptoms appear [104]. Non-invasive tests analyzing gut microbiota composition could provide valuable diagnostic tools, aiding in the timely and accurate identification of individuals at risk for AD [105].

The interplay between gut microbiota and neuroinflammation underscores the importance of personalized medicine in AD treatment [106]. Individuals have unique microbiota compositions influenced by genetics, diet, lifestyle, and environmental factors [107]. Personalized therapeutic strategies targeting the gut–brain axis could be developed to optimize treatment efficacy for each patient [108]. By tailoring interventions based on an individual’s specific microbiota profile, better clinical outcomes may be possible [109].

Utilizing multi-omics approaches, including genomics, transcriptomics, proteomics, and metabolomics, can provide a comprehensive understanding of the gut–brain axis in AD [110]. These approaches can identify key microbial and host factors in disease progression and intervention response. Integrating multi-omics data with clinical and cognitive assessments can lead to the discovery of novel biomarkers and therapeutic targets [111].

### 3.2. Nutraceuticals and Dietary Modulation of the Microbiome: Emerging Mechanisms in Alzheimer’s Disease

Recent studies have revealed new avenues through which nutraceuticals and dietary interventions can influence the gut microbiome and neuroinflammation in Alzheimer’s disease (AD). While much attention has been given to well-known components, such as short-chain fatty acids (SCFAs) and probiotics, recent research highlights underexplored mechanisms and compounds that could provide novel therapeutic strategies for AD [112].

Quercetin, a flavonoid abundant in foods like apples, berries, and onions, has recently gained attention for its potent microbiome-modulating properties [113]. Quercetin has demonstrated a unique ability to selectively enhance beneficial gut bacteria, such as Bifidobacterium and Lactobacillus, unlike more commonly studied polyphenols, while reducing pathogenic bacteria linked to gut dysbiosis [114]. These shifts in the microbiome promote the production of anti-inflammatory SCFAs and improve gut barrier integrity, reducing the systemic translocation of pro-inflammatory molecules like lipopolysaccharides (LPS), which are implicated in the activation of neuroinflammatory pathways in AD. Early animal studies suggest that quercetin’s microbiome modulation could reduce neuroinflammatory markers in the brain, making it a promising candidate for further investigation in AD research [115].

Curcumin, a bioactive compound found in turmeric, has also emerged as a powerful modulator of the gut microbiome with potential implications for neuroinflammation in AD. Recent research suggests that curcumin enhances the population of butyrate-producing bacteria in the gut, such as *Faecalibacterium prausnitzii*, which plays a critical role in reducing systemic inflammation and preserving the integrity of the BBB. By improving microbial composition, curcumin may decrease the levels of pro-inflammatory cytokines in the brain, particularly those that activate microglial cells, which are central to AD pathology. This indicates a dual action of curcumin—directly reducing neuroinflammation while simultaneously restoring gut homeostasis [116].

While much attention has been focused on SCFAs, bile acids are another group of microbiota-derived metabolites that have recently been linked to neuroinflammation in AD. Bile acids, synthesized in the liver and metabolized by gut bacteria, play a crucial role in regulating systemic inflammation through their interaction with the farnesoid X receptor (FXR) and Takeda G-protein-coupled receptor 5 (TGR5). Dysbiosis in AD patients often leads to imbalances in bile acid metabolism, which can exacerbate systemic inflammation and contribute to the breakdown of the BBB [117]. Novel studies suggest that restoring the balance of bile acids through dietary or microbial interventions could reduce peripheral inflammation and neuroinflammatory responses in the brain, opening new pathways for therapeutic exploration [118]. Certain secondary bile acids produced by gut bacteria have been shown to modulate immune responses within the CNS. These findings highlight a potential link between gut-derived bile acids and microglial activation, suggesting that manipulating bile acid metabolism via the gut microbiota could offer a new method for controlling neuroinflammation in AD [119].

Polyunsaturated fatty acids (PUFAs), such as omega-3 (EPA and DHA) and omega-6 fatty acids, are well-known for their anti-inflammatory properties. Still, recent research has begun to explore how these compounds interact with the gut microbiome to exert their effects on brain health. Omega-3 fatty acids, in particular, have been shown to promote the growth of specific beneficial bacteria that produce anti-inflammatory metabolites, including SCFAs and certain lipid mediators, which can cross the BBB and modulate microglial activity. This interaction between PUFAs and the gut microbiome is thought to amplify their neuroprotective effects, helping to resolve chronic inflammation associated with AD [120].

### 3.3. Gender Differences in Microbiota and Relevance to Neuroinflammation in Alzheimer’s Disease

Males and females exhibit distinct gut microbiota profiles, which may influence the susceptibility to neuroinflammatory conditions, including AD [121]. These differences are influenced by various factors, including sex hormones (e.g., estrogen and testosterone), immune function, and metabolic processes. For instance, estrogen has been shown to promote the growth of beneficial bacterial species, such as *Lactobacillus* and *Bifidobacterium*, which produce anti-inflammatory metabolites like SCFAs [122].

The interplay between the gut microbiota and the immune system is critical in AD. Dysbiosis, or an imbalance in gut microbiota, produces pro-inflammatory molecules such as LPS, which can cross the BBB and activate microglial cells. In males, more pro-inflammatory gut bacteria may activate these neuroinflammatory pathways more pronounced, accelerating neuronal damage and cognitive decline [89,123].

Males tend to have a higher abundance of pro-inflammatory bacterial species associated with increased intestinal permeability and the translocation of bacterial endotoxins, such as lipopolysaccharides (LPS). This difference may contribute to a higher risk of chronic systemic inflammation and neuroinflammation in males, potentially exacerbating the progression of AD. These gender-specific differences in microbial composition suggest that males may be more susceptible to LPS-induced neuroinflammatory pathways, which activate microglial cells and contribute to the accumulation of amyloid-beta (Aβ) plaques in the brain [124].

After menopause, the decline in estrogen levels may lead to shifts in gut microbiota composition, increasing susceptibility to systemic inflammation and neuroinflammation. This hormonal change is associated with a reduction in beneficial bacteria and an increase in pro-inflammatory species, which can exacerbate gut permeability and promote the translocation of LPS into the bloodstream. In turn, this triggers neuroinflammatory responses in the brain, contributing to the progression of AD. This post-menopausal shift highlights the importance of considering hormonal status when evaluating microbiota-targeted therapies in women with AD [125].

Testosterone has been shown to modulate immune function and microbial composition in males, with complex effects on inflammation. While testosterone may have protective effects against some forms of systemic inflammation, it is associated with an increased prevalence of pro-inflammatory gut bacteria. This suggests that testosterone may not provide the same level of neuroprotective benefits as estrogen, potentially contributing to the observed gender differences in AD risk and progression [126].

Immune system differences between males and females also contribute to the observed gender-specific effects of the gut microbiota on neuroinflammation. Females generally exhibit stronger immune responses, characterized by a more robust production of pro-inflammatory and anti-inflammatory cytokines. This heightened immune activity may help to clear pathogens and reduce systemic inflammation in younger females. However, it also makes them more susceptible to autoimmune disorders and chronic inflammation in later life, particularly after menopause [127].

Microbiota-based interventions, such as probiotics, prebiotics, and dietary modifications, may need to be tailored according to gender to optimize their effectiveness. For instance, probiotic strains that promote the growth of SCFA-producing bacteria may be particularly beneficial for post-menopausal women, helping to restore gut barrier integrity and reduce systemic inflammation [128].

Personalized nutrition and microbiota-targeted therapies based on individual microbiota profiles may offer a more effective way to address the complex interplay between gender, microbiota, and neuroinflammation in AD [129]. Future research should focus on identifying specific bacterial species and metabolites differentially regulated in males and females, allowing for more precise therapeutic interventions.

## 4. Experimental and Clinical Studies on Microbiota and Neuroinflammation in AD

Numerous animal studies have explored the impact of gut microbiota on neuroinflammation and AD pathology, revealing significant insights into the microbiota–neuroinflammation connection [130,131]. Studies using germ-free mice or mice treated with antibiotics to deplete gut microbiota have demonstrated that these mice exhibit reduced amyloid-beta (Aβ) deposition and neuroinflammation, emphasizing the microbiota’s role in AD [132,133]. Upon recolonization with specific bacterial strains, these pathological features can be restored, underscoring the critical influence of gut microbiota [134]. Probiotic and prebiotic interventions in AD animal models further support this connection [135]. For instance, supplementation with Lactobacillus and Bifidobacterium strains has been shown to reduce neuroinflammation, improve gut barrier function, and enhance cognitive performance [136]. These treatments are associated with decreased microglial activation and lower levels of pro-inflammatory cytokines, highlighting the therapeutic potential of targeting gut microbiota to mitigate AD symptoms and progression [137].

Clinical studies have provided complementary insights by investigating the gut microbiota composition and potential therapeutic interventions in AD patients [138]. Cross-sectional studies comparing gut microbiota between AD patients and healthy controls have consistently reported dysbiosis, characterized by alterations in specific bacterial taxa and reduced microbial diversity. These findings suggest a link between gut microbiota imbalances and AD pathology [139].

Longitudinal studies tracking changes in gut microbiota over time in AD patients suggest that microbiota alterations may precede clinical symptoms, indicating a potential role in disease onset and progression [140]. Intervention trials have further explored this potential, examining the effects of probiotics, prebiotics, and dietary changes on AD patients [141]. Some clinical trials have shown promising results, such as improved cognitive function and reduced inflammatory markers following probiotic supplementation [142,143]. However, the results are variable, and larger, more extensive trials are needed to confirm these findings and establish robust clinical guidelines for microbiota-targeted therapies in AD [144].

Recent studies have provided substantial evidence supporting the gut–brain axis’s involvement in AD [145]. Significant differences in the gut microbiota composition of AD patients were reported compared to cognitively normal controls, with a notable increase in pro-inflammatory bacteria and a decrease in short-chain fatty acid (SCFA)-producing bacteria [146]. Harach et al. (2017) conducted a pivotal study using an AD mouse model, where germ-free mice exhibited reduced Aβ pathology, which was reversed upon microbial recolonization, demonstrating the direct impact of gut microbiota on AD pathology [147]. Kim et al. (2021) conducted a clinical trial that showed a 12-week probiotic intervention in AD patients resulted in improved cognitive function and reduced serum levels of pro-inflammatory cytokines, suggesting the potential benefits of probiotics in managing AD [148]. These studies collectively highlight the significant role of gut microbiota in modulating neuroinflammation and AD pathology, opening new avenues for therapeutic interventions targeting gut microbiota to ameliorate neuroinflammation and slow AD progression [149]. Further research is essential to validate these findings through longitudinal studies and clinical trials, ultimately developing effective microbiota-based therapies for AD.

## 5. Discussion

This review explores the intricate relationship between neuroinflammation, microbiota, and AD, revealing several critical findings. Neuroinflammation is a central feature of AD, characterized by the activation of microglia and astrocytes, which leads to the release of pro-inflammatory cytokines and chemokines [150]. This chronic inflammatory response exacerbates neuronal damage and accelerates AD progression [151]. Furthermore, AD patients exhibit significant alterations in gut microbiota composition, including reduced microbial diversity and an increased abundance of pro-inflammatory bacteria [152]. Beneficial bacteria that produce anti-inflammatory metabolites are often diminished in these patients, suggesting a vital role for the gut microbiota in the pathology of AD [153].

Dysbiosis, or microbial imbalance, can increase intestinal permeability, allowing bacterial endotoxins such as lipopolysaccharides (LPS) to enter the bloodstream and cross the BBB, triggering neuroinflammatory responses [154]. Microbial metabolites, such as SCFAs, play crucial roles in modulating inflammation and maintaining brain health [155]. Animal studies [156] and clinical trials provide compelling evidence that modulating the gut microbiota can influence neuroinflammation and cognitive outcomes in AD [157]. Probiotic and prebiotic interventions have shown potential in reducing neuroinflammation and improving cognitive function, suggesting that targeting the gut microbiota could be a promising therapeutic strategy for AD [158].

This review highlights several mechanisms through which gut microbiota influence neuroinflammation in AD. Dysbiosis-induced increased intestinal permeability allows endotoxins like LPS to enter the bloodstream, promoting systemic inflammation [159]. These endotoxins can cross the BBB and activate microglia, leading to sustained neuroinflammation [160]. The gut microbiota also modulates immune responses, with dysbiosis leading to heightened systemic inflammation [161]. Pro-inflammatory cytokines produced in the periphery can affect the central nervous system (CNS), exacerbating neuroinflammation in AD [162]. Additionally, beneficial bacteria produce SCFAs, which have anti-inflammatory properties and can modulate microglial activation [163]. Dysbiosis reduces SCFA production, contributing to a pro-inflammatory environment in the brain [164].

This review provides a detailed analysis of the relationship between gut microbiota and neuroinflammation in AD (Figure 1), integrating findings from both experimental and clinical studies. This review offers a multidisciplinary perspective on AD pathology by considering various aspects of microbiota and neuroinflammation. The potential of microbiota-targeted therapies is highlighted, paving the way for future clinical applications. However, this review also acknowledges several limitations. Variability in study designs, populations, and methodologies across the included studies may limit the generalizability of the findings. Many studies are cross-sectional, providing only a snapshot of microbiota composition and neuroinflammation at a single time point. Longitudinal studies are needed to understand causal relationships.

Differences in microbiota sampling, sequencing techniques, and data analysis methods can affect the comparability of results across studies. Additionally, the review may be subject to publication bias, with positive findings more likely to be published than negative or null results. Future research should focus on several areas to advance our understanding of microbiota’s role in neuroinflammation and AD. Longitudinal studies are essential to track changes in gut microbiota composition and neuroinflammatory markers over time in AD patients, from preclinical to advanced stages. Mechanistic studies should investigate the molecular mechanisms by which gut microbiota influences neuroinflammation, focusing on pathways such as the gut–brain axis, immune modulation, and microbial metabolite production.

Interventional trials should be designed and conducted to evaluate the efficacy of microbiota-targeted interventions, such as probiotics, prebiotics, and dietary modifications, in reducing neuroinflammation and improving cognitive outcomes in AD patients. Employing multi-omics approaches, including genomics, transcriptomics, proteomics, and metabolomics, can provide a comprehensive understanding of the interactions between gut microbiota and host physiology in AD. Developing personalized therapeutic strategies based on individual microbiota profiles can optimize treatment efficacy and minimize adverse effects. Finally, integrating microbiota research with established AD biomarkers, such as amyloid-beta (Aβ) and tau, can enhance our understanding of the interactions between peripheral and central pathologies, potentially leading to more effective diagnostic and therapeutic strategies for AD.

## 6. Conclusions

This review highlights the significant role of neuroinflammation and gut microbiota in the pathogenesis of AD. Alterations in gut microbiota composition could serve as early biomarkers for AD, potentially allowing for earlier diagnosis and intervention. Non-invasive microbiota profiling could be integrated into clinical practice to identify individuals at risk for AD. Therapeutic strategies to modulate the gut microbiota offer a novel approach to mitigating neuroinflammation and slowing AD progression. Probiotics, prebiotics, and dietary interventions should be further explored and validated in clinical trials to determine their efficacy and safety in AD patients. The variability in microbiota composition among individuals underscores the need for personalized therapeutic approaches. Tailoring interventions based on each patient’s unique microbiota profile could enhance treatment efficacy and minimize adverse effects.

In conclusion, this review underscores the critical interplay between neuroinflammation and gut microbiota in AD. Understanding and modulating this relationship holds significant promise for developing innovative diagnostic and therapeutic strategies, ultimately improving outcomes for individuals affected by this debilitating disorder. Continued research and clinical efforts are essential to translating these findings into practical applications and enhancing AD patients’ and their families’ quality of life.

## Figures and Tables

**Figure 1 ijms-25-11272-f001:**
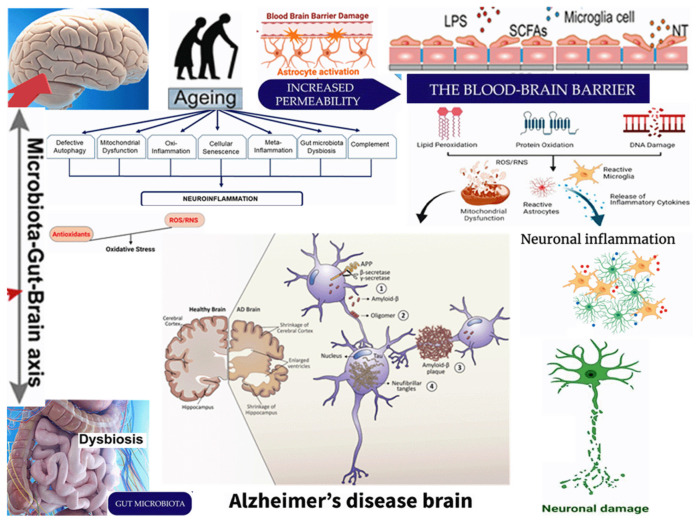
The intricate relationship between neuroinflammation, microbiota, and Alzheimer’s disease.

**Table 1 ijms-25-11272-t001:** Search terms used for studies focusing on neuroinflammation and microbiota in AD.

Keywords	Elsevier	PubMed	PMC	PEDro	Total
“Alzheimer’s disease” + “Neuroinflammation” + “Autophagy” + “Microbiota”	3	8	2	0	13
“Alzheimer’s disease” + “Neuroinflammation” + “Apoptosis” + “Microbiota”	0	7	4	0	11
“Alzheimer’s disease” + “Neuroinflammation” + “Neuroplasticity” + “Microbiota”	0	2	0	0	2
“Alzheimer’s disease” + “Neuroinflammation” + “connectomics” + “Microbiota”	0	0	0	0	0
“Alzheimer’s disease” + “Neuroinflammation” + “Circuitry” + “Microbiota”	0	0	0	0	0
“Alzheimer’s disease” + “Neuroinflammation” + “Mitophagy” + “Microbiota”	0	0	0	0	0
“Alzheimer’s disease” + “Neuroinflammation” + “Oxidative Stress” + “Microbiota”	7	26	13	0	46

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
