# Peer review of "Novelties on Neuroinflammation in Alzheimer’s Disease–Focus on Gut and Oral Microbiota Involvement"

_ijms, 2024, doi:10.3390/ijms252011272_

Round 1
Reviewer 1 Report
Comments and Suggestions for Authors
In the systematic review entitled “Novelties on neuroinflammation in Alzheimer's disease - focus on microbiota involvement. The Systematic review” by Popescu et al. highlighted the critical role of the microbiota in AD pathology and also offered a new insight that modulation of the gut and oral microbiota could be a new therapeutic strategy to reduce neuroinflammation and slow disease progression.
Major revisions
- The authors have dealt with a topical and interesting subject but should have expanded on some salient points. The systematic review does not add much to what is already in the literature and is incomplete in some aspects that should be expanded to give readers a broader and more detailed overview of the problem. For example, it is important to clarify how nutraceuticals and diet can impact changes in the microbiome and the production of SCFAs. Many reviews and studies have investigated the role of micronutrients, pro/prebiotics, nutraceuticals (e.g. Andrographis), or diet in general on neuroinflammation and AD. Therefore, this gap should be filled by including such information as well.
- Another finding that could make the manuscript more innovative could be the gender difference in the microbiota variations underlying neuroinflammation and AD.
Minor Revisions
- To make the title more specific and direct instead of ‘Novelties on neuroinflammation in Alzheimer's disease - focus on microbiota involvement. Systematic review' it would be more appropriate to use “Novelties on neuroinflammation in Alzheimer's disease - focus on gut and oral microbiota involvement: a systematic review”.
- Arrange all the graphic appearance of Figure 1 (e.g. arrows.)
- Check that all bibliographic entries are placed in the correct place in the text (e.g. bibliographic entry 63 at the end of the sentence)
Comments on the Quality of English LanguageMinor editing of English language required
Author Response
Dear Reviewer,
We appreciate your time and effort in reviewing our manuscript, “Novelties on neuroinflammation in Alzheimer's disease—focus on microbiota involvement.” We are grateful for your valuable feedback and have carefully addressed all your comments and suggestions.
Major Revisions:
-
We agree with your observation that more information should be included about the role of nutraceuticals and diet in modulating the microbiome and its relevance to Alzheimer’s disease (AD) pathology. We have expanded the discussion to include the impact of micronutrients, pro/prebiotics, and specific nutraceuticals on the microbiome, emphasizing their role in altering short-chain fatty acids (SCFAs) production and neuroinflammation.
-
We agree that gender-specific differences in microbiota composition and neuroinflammation in AD are essential to consider. We have included a new subsection in the Discussion highlighting current evidence on gender differences in microbiota changes and their implications for AD progression.
Minor Revisions:
-
We have modified the title to more explicitly reflect the involvement of both gut and oral microbiota. The revised title is: “Novelties on neuroinflammation in Alzheimer's disease - focus on gut and oral microbiota involvement.”
-
We have revised the graphical elements in Figure 1 to improve clarity and visual appeal, including adjustments to the arrows and overall layout. We believe the figure now presents the information more clearly.
-
We carefully reviewed all the bibliographic entries to ensure they are correctly placed. Specifically, we corrected the placement of entry 63 and others that required adjustments.
Reviewer 2 Report
Comments and Suggestions for Authors
I found the topic of the article very interesting and relevant. Authors seem to reached a lot of valuable text. But I have a big problem with this manuscript. First of all there is no clear hypothesis to be tested through the literature search. And the second thing: the review is a type of a narrative review, very old fashioned. I would suggest to more methodological approach to present data from articles. There is no list of articles included, there is no analysis of data extracted from articles, only plain narration, good for introduction and discussion.
Author Response
Dear Reviewer,
Thank you for your thoughtful feedback and for acknowledging the topic's relevance. We are grateful for your suggestions, which have led us to reconsider our manuscript's structural and methodological aspects.
We understand your concern regarding the absence of a clearly defined hypothesis. To address this, we have restructured the manuscript to articulate a straightforward research question and hypothesis. Specifically, we have now clarified that the primary aim of the systematic review is to investigate the hypothesis that alterations in gut and oral microbiota contribute to neuroinflammation, which plays a critical role in the progression of Alzheimer’s disease (AD).
We have included a PRISMA (Preferred Reporting Items for Systematic Reviews and Meta-Analyses) flow diagram to provide transparency in the article selection process. This outlines the number of articles retrieved, screened, excluded, and included in the final review.
In response to your suggestion, we have provided a table listing all the articles found by the systematic review. We have revised the Results section to include a more detailed analysis of the extracted data from the included studies. The discussion has been refined to integrate the results of the included studies.
Reviewer 3 Report
Comments and Suggestions for Authors
In the manuscript (ID: 265151) by Popescu et al., minor revisions are recommended. Although the work is overall well-structured and the topics discussed are of significant scientific interest, certain sections could benefit from slight improvements. Specifically, it is advised to correct any typographical errors and revise the structure of some sentences to enhance clarity and readability. Moreover, greater attention to the journal’s stylistic guidelines, particularly regarding punctuation and citation formatting, would contribute to improving the overall quality of the manuscript. These adjustments do not affect the scientific content but ensure that the manuscript fully complies with the required editorial standards. In conclusion, these are minor corrections that, once implemented, should facilitate the final acceptance of the work without the need for further revisions.
The text contains some minor typographical errors that need to be corrected. Additionally, the journal's author guidelines specify that punctuation should be placed after the square brackets containing the reference numbers. I am citing the relevant information from the author instructions available at the following link: https://www.mdpi.com/journal/ijms/instructions. Adhering to these formal guidelines is essential to ensure that the manuscript complies with the journal's editorial standards and to avoid potential revisions or modification requests from the reviewers or editor.
The introduction of the article is well-written and presents the content in a clear and effective manner. However, to further enhance readability and make the text more engaging, it would be beneficial to divide it into two or three distinct paragraphs. This approach would not only help in better organizing the main ideas, but also allow the reader to absorb the information in a more gradual and natural way. Proper paragraph division creates visual pauses that facilitate comprehension and help the reader follow the logical flow with greater ease. Additionally, incorporating an image could further lighten the reading experience, making the text visually more dynamic and appealing. Images can provide a visual context to the information, aiding the reader in better understanding the content and maintaining their attention. In an era where readers' attention spans are increasingly limited, these strategies are essential for keeping the reader engaged and improving the overall reading experience. In summary, a more segmented structure and the thoughtful use of images would not only make the article more readable but also more engaging and accessible, without compromising the quality of the content.
Acronyms should be written out in full only the first time they are mentioned, after which either the acronym or the full term should be used consistently. However, there is some inconsistency in the current text: in some sections, the acronym is followed by its explanation, while in others, only the acronym is used without following a uniform rule. For example, in lines 243 and 254, the use of acronyms does not follow the same pattern. It is important to standardize the use of acronyms to improve the consistency and readability of the manuscript.
Author Response
Dear Reviewer,
We greatly appreciate your time and the insightful feedback you have provided on our manuscript "Novelties on neuroinflammation in Alzheimer's disease - focus on microbiota involvement. A Systematic Review." Your suggestions are valuable, and we have carefully considered each point.
We thoroughly reviewed the entire manuscript and corrected the typographical errors you identified. Additionally, several sentences were restructured to improve clarity and readability. We ensured that punctuation was placed after the square brackets containing reference numbers. We also reviewed the placement of all citations to ensure consistency and correctness throughout the text. The flow of information is now more logical and accessible to follow, with clear transitions between the main points.
We addressed the inconsistency in the use of acronyms throughout the manuscript. All acronyms are now written out in full upon their first mention, and their use has been standardized in the following instances. For example, issues with acronyms in lines 243 and 254 (as noted in your feedback) have been corrected to follow a uniform pattern. This change enhances the manuscript’s coherence and readability.
Round 2
Reviewer 2 Report
Comments and Suggestions for Authors
The article is much better, good work